# Proteomic Approaches in the Study of Placenta of Pregnancy Complicated by Gestational Diabetes Mellitus

**DOI:** 10.3390/biomedicines10092272

**Published:** 2022-09-14

**Authors:** Annunziata Lapolla, Pietro Traldi

**Affiliations:** 1Department of Medicine, University of Padova, 35122 Padova, Italy; 2Istituto di Ricerca Pediatrica, Città della Speranza, 35127 Padova, Italy

**Keywords:** gestational diabetes, placenta, mass spectrometry, 2DGel Electrophoresis, MALDI imaging

## Abstract

Gestational diabetes mellitus (GDM), a glucose intolerance developing or first recognized during pregnancy, leads to a series of short- and long-term maternal and fetal complications, somehow related to placenta structural and functional changes. The focus and the objective of the present review are to discuss the results which can be obtained by different mass spectrometric approaches in the study of placenta protein profile. Thus, matrix-assisted laser desorption/ionization mass spectrometry (MALDI) has been applied on placenta omogenates before and after one-dimensional electrophoretic separation, followed by tryptic digestion. MALDI imaging was used for direct investigation on the placenta tissue (both maternal and fetal sides). The results showed that some differences among the absolute abundances of some proteins are present for placenta samples from GDM patients. The majority of investigations were carried out by two-dimensional electrophoresis (2DE) followed by LC-MS/MS or, directly by the label-free LC-MS^E^ approach. It should be emphasized that all these techniques were showed differences in the protein expression between the placenta samples from healthy or GDM subjects. 2DE was also employed to separate and compare placental protein levels from GDM and the control groups: differentially expressed proteins between the two groups were identified by MALDI-TOF/TOF mass spectrometry and were further confirmed by Western blotting. The physiopathological significance of the obtained results are reported and discussed in this narrative review. The experimental data obtained until now show that the newest, mass spectrometric approaches can be considered a valid tool to investigate the possible changes of placenta in the presence of GDM.

## 1. Introduction

As an introduction we consider it essential to describe the biological and medical aspects of the placental pathophysiologies. The placenta has a key role in correct fetal development. A series of factors are related to the correct functioning of the placenta as maternal and fetal blood flow, expression and function of receptors and transporters, and appropriate nutrients. It is well known that placenta in diabetes undergo a series of structural and functional changes due to the abnormal maternal milieu determined by elevated levels of glucose ad insulin [1]. Gestational diabetes mellitus (GDM), a glucose intolerance developing (or first recognized) during pregnancy that is not an overt diabetes [2], has been increasing over the last decade. It accounts for 12–18% of all pregnancies and is due mainly to the increased frequency of obesity [2,3]. This condition, if not properly diagnosed and treated, determines a series of short- and long-term maternal and fetal complications such as preeclampsia, cesarean delivery, birth trauma, macrosomia, neonatal hypoglycemia and hyperbilirubinemia [3,4,5]. Furthermore, women affected by GDM and their children are at high risk of developing cardiometabolic diseases (including type 2 diabetes, obesity, hyperlipemia, metabolic syndrome, hypertension, and cardiovascular disease) later in life [4,5]. GDM develops when beta cell insulin secretion is unable to compensate for the physiological pregnancy-induced insulin resistance, and/or in conjunction with an impaired beta cell function [6,7]. In this context, as recently evidenced, genetic, epigenetic and environmental factors contribute in determining insulin resistance and beta-cell disfunction, and consequently GDM development. In addition, the adverse intrauterine environment in patients with GDM could also have a negative impact on the establishment of the epigenomes of the offspring [6,7,8]. Furthermore, GDM is associated with altered concentrations of nutrients, inflammatory cytokines which can contribute to placenta modifications. This includes changes in the surface area and volume, as well as histological changes as an increased volume of the intervillous space and terminal villi, the number of syncytiotrophoblast, fibrinoid and glycogen deposits. These modifications may result in functional changes of the placenta, with consequent impairment of fetal development [1]. Recently, the new omics methodologies have been retained, as they are useful for the identification of new pathways and processes that could be affected in placenta modified by diabetes [9]. In this context, proteomics, based on the structural identification and quantitative evaluation of the proteins present in a cell, tissue or organism in a well-defined moment, and related to pathological conditions, has been proved to be a highly valid approach, together with metabolomics, genomic and metallomics [10]. It must be considered that the changes of the various biochemical pathways characteristic of an organism originating by a specific disease, drug or physiological activity reflect in the identification of novel biomarkers, is useful in the management of the diseases in clinical practice [11]. The identification of modified proteins in placenta of GDM patients and their possible involvement in placenta and fetal development is of key importance in order to outline prevention strategies of maternal and fetal complications. In a review focused on proteomic studies on placenta samples and placenta-derived cells of normal pregnant women, Robinson et al. [12] concluded that proteomic analysis is a promising approach to obtain information that is difficult to obtain with traditional approaches in pathological conditions. It should be emphasized that different instrumental approaches available nowadays have been successfully applied, and in this narrative review the results obtained by different research groups are described and discussed.

## 2. Placenta Protein Profile

In order to evaluate the results obtained in the study on the placenta protein profile, we considered it of interest to employ a general strategy based on four different aspects: (i) samples analyzed; (ii) instrumental approaches employed; (iii) obtained results; (iv) physiopathological meaning of the results. A preliminary investigation on the placenta protein profile [13] was based on pooled placenta omogenates from 20 healthy subjects and 20 GDM pregnant women.

The analytical strategy employed in the study is summarized in Figure 1. The biological samples under study were obtained by removing the placental cotyledon from the central region of the placenta to obtain villous tissue. Once thawed, 1 g of each placenta sample was washed with distilled water, dipped in ice-cold homogenization buffer and homogenized under ice-cooling with a blender. The homogenate was centrifuged twice at 15,000 rpm for 10 min. The supernatant, containing the hydrosoluble proteins, was recovered, collected and pooled in two groups: “control” and “diabetic”. The two pooling samples were directly analyzed by matrix-assisted laser desorption/ionization mass spectrometry (MALDI MS) [14], and the proteins detected were tentatively identified by comparison of their molecular weight with the Human Protein Reference Database, restricting the search to the species expressed in placenta tissue. At first sight this method seemed to lead to satisfactory results, with the detection in the *m*/*z* range 15,000–32,000 of: 39S ribosomal protein I.55, insulin–like peptide INSI.5, ergosterol biosynthetic protein 28, histone H3-like centromeric protein A, R-spondin-3 and mitocondrial dicarboxylate carrier. However, an in-depth analysis of these results showed the presence of relevant limits of this approach: the most abundant ionic species in the MALDI spectra were proved to be due to globins present in the maternal and fetal blood. Furthermore, an evaluation with MS/MS of the placenta samples omogenates subjected to trypsin digestion confirms this result. These findings focalize the attention that needs to be paid to avoid blood contamination of the placenta samples, and in the analysis of the data obtained from database comparison. The one-dimensional electrophoretic separation of the proteins present in the two pools was performed, followed by tryptic digestion of the proteins evidenced by the different bands (right part of Figure 1). With this accurate method, a large number of proteins in the mass range 15–220 kDa were found and, after digestion with trypsin, MALDI analysis made possible to identify a series of proteins (different from globins), with coverage values ranging from 34% to 78%. Densitometric analysis indicated a slightly higher optical density in the bands. This was due to a serine/arginine repetitive matrix protein 1 and Bcl2-associated transcriptor factor1 for the GDM pool with respect to the same bands in the control pool. These structural assignments were confirmed by MALDI MS/MS experiments, which in the case of samples from the diabetic pool, show also the presence of glyco-oxidized species, i.e., the products of non-ezymatic glycation and glycol-oxidation arising from the reaction of glucose with amino groups of proteins. It was noted that only minor quantitative differences were found between proteins for GDM and normal pregnancies; these data can be explained by the fact that the GDM women evaluated were in good metabolic control, showing only slightly elevated HbA1c values with respect to normal pregnant women. It could be emphasized that the analysis of a pool sample exhibits some limitations, giving a general view of the possible differences but lacking the aspects related to single subject.

To evaluate the role of the placenta sample treatments on the achievable results, it was considered of interest to conduct an investigation on the placenta protein profile. This was carried out by matrix-assisted laser desorption/ionization ion imaging [15] experiments, performed by direct laser irradiation of the maternal and fetal sides of the placenta tissue [16]. To investigate the possible GDM-induced changes, the results obtained for five placenta samples from GDM patients were compared with those from five placenta samples of healthy pregnant women. This investigation required an in-depth evaluation of the sample treatments for the MALDI ion imaging operative conditions: four different operative protocols were tested and the stainless–steel sieve method [17] for matrix deposition proved to be particularly effective. However, as already observed in the MALDI spectra of placenta omogenates, the most abundant peaks obtained by direct irradiation of the placenta tissue are due to blood globins, even after the sequential washing procedure (up to five). This behavior was also observed by Wang et al. [18] in screening multi-protein complexes in placenta: heterooligomeric multi-protein complexes (including mitochondrial respiratory chain, integrin, proteasome, histone, and heat shock protein complexes) were identified. These results were obtained by utilizing an electrophoresis method (BN SDS-PAGE) coupled with a mass spectrometry approach (LC-MS/MS), i.e., in experimental conditions leading to results more specific with respect to those achievable by MALDI. Attention was then focused on the species detected in the *m*/*z* range 20,000–47,000. The average MALDI ion imaging spectra, obtained for two placenta samples taken as an example, evidenced differences in the absolute abundances of the ionic species at *m*/*z* 30.335, *m*/*z* 31,235 and *m*/*z* 32,000, due to NADH dehydrogenase iron-sulfur protein 3, mitochondrial, transcription cofactor vestigial-like protein 4, voltage dependent anion-selective channel protein 2 respectively. An interesting result was observed for the ion at *m*/*z* 31,325 (transcription cofactor vestigial-like protein 4): it is the most abundant species only in the case of the maternal placenta side of the GDM subject, indicating that changes in the placenta protein profile in GDM patients are present mainly in the maternal side of the placenta, and that the placenta action seems to inhibit these changes on the fetal side. However, the low number of patients under study must be stressed, indicating the necessity of a large number of healthy and GDM subjects. When these data were evaluated in the 20 placenta samples under investigation, the differences above discussed are not so evident. The ion intensity from the average spectra of the different samples show that clear differences are present for the ions at *m*/*z* 30,335 and 31,235 between the maternal and fetal sides, both for healthy and the GDM pregnant women. It was noted that many of the detected proteins are present at the mitochondrial These preliminary results seemed to be very promising as evidence of differences at the protein level between maternal and fetal sides of the placenta. However, in our opinion, in order to confirm these findings and to evaluate their physiopathological meaning, further work is required on a larger number of placenta samples. Only after this in-depth further analyses it would be possible to evaluate which placental side should be used in future experiments.

The above-described approaches exhibit severe limitations. Consequently, it was considered interesting to utilize the classical and more specific approaches employed in proteomics, i.e., 2D gel-based electrophoresis followed by nanoLC-MS/MS or MALDI-MS/MS and MS^E^ (gel free) analysis of the different spots. The data obtained by 2D gel-based and gel-free proteomic approaches on human placental tissue of GDM patients were then compared [19]. The gel-based approach highlights, in the case of GDM samples, 13 over-expressed proteins and 16 under-expressed proteins. This underlines that, while nanoLC-MS/MS shows that electrophoresis spots contain more than one protein, MALDI-MS/MS leads to the identification of only one protein per spot, indicating the highest effectiveness of the former analytical approach. The gel-free approach was successively applied to placenta tissues, and the enzymatic digestion products of the whole placental tissue were analyzed by a label-free LC-MS^E^ method. This method allows not only the identification but also the quantification of proteins. The results obtained can be ascribed to a reproducible chromatographic separation followed by the MS/MS analysis, performed by alternating acquisition of product ion spectra at lower and higher collision energies. This approach led to the detection of 159 proteins, of which 10 over-expressed in GDM placental tissue and 9 under expressed compared to normal placenta: Periostin, Ig gamma-2 chain C region, CSH, Moesin, Heat-shock-related 70 kDa protein 2, triosephospate isomerase, Protein disulfide-isomerase, Galectin-1, Vimentin, 14-3-3 protein beta/alpha. Some of these proteins were identified also in the analysis performed with the gel-based approach.

Differently to what was performed in previous investigations, based on the pooling of placenta samples, in a further study [20] each placenta sample obtained from healthy (12) and GDM (13) subjects was analyzed individually by means of the label-free LC-MS^E^ method. This approach allowed us to identify 3776 peptides, corresponding to 160 proteins; statistical analysis allowed us to highlight higher levels of galectin 1 and collagen alpha-1 chain in the case of GDM samples, while heat shock protein 1A/1B was less abundant in the GDM placental tissue. Galectin-1 is a carbohydrate-binding protein with affinity for beta-galactosides, which in pregnancy has been implicated in regulating processes associated with adaptation to pregnancy and in mechanisms involved in angiogenesis, trophoblast invasion, and syncytium formation. Its dysregulation has been associated with adverse pregnancy outcomes as spontaneous abortion, preeclampsia, and HELLP syndrome. In a recent paper evaluating placental tissues with dual immunofluorescence, Blois et al. [21] showed that serum galectin-1 levels were reduced, while placental galectin-1 was overexpressed in GDM women. Furthermore, a slightly higher abundance of collagen alpha-1 chain was also found in placenta samples of GDM women.

It is noted that clear differences are present among the “diagnostic” proteins when considering the results of the study of Burlina et al. [20] and those identified in the previous studies based on pooled placenta samples. In the latter case, a general view of the protein map, due to proteins present in different samples, is obtained. On the contrary, the findings on specific placenta samples indicate that the placental proteome is weakly affected by GDM. This is different to what was observed for metabolome, which exhibits major changes in the presence of GDM. The placenta proteome appears to be scarcely modified by GDM, and this can be due to the good glycemic control of the patients evaluated; furthermore, the abnormalities put in evidence are the view of GDM at diagnosis, when patients are not treated with dietetic advices and/or insulin.

Favaro et al. [22] studied the extracellular matrix remodeling in a mouse model of pregnancy complicated by type 1 diabetes in bad metabolic control, and a significantly increased deposition of type I collagen in the decidua was found. These results are in line with the findings obtained in human placenta samples, but the low abundance increase of collagen alpha-1 XIV chain in the case of GDM placenta samples can be related to the good metabolic control of GDM women evaluated. Also, HSP70 (a stress-induced protein involved in protein folding, degradation, and transport) was found in slightly increased abundance in the healthy control group. The increased levels of HSP70 are determined by the increase of inflammation and oxidative stress evidenced in type 1 and 2 diabetes, and in GDM that shares a common physiopathology of type 2 diabetes. In this context, the paper of Burlina et al. [20] has evidenced, for the first time, a higher abundance of HSP70 in the placenta of healthy controls with respect to women with GDM. Abdulsid et al. [23], in their paper on placenta samples of normal pregnant women and women that undergo preeclampsia complication, showed a significant increase of the placental expression of HSP70 during labor, in comparison with delivery via cesarean section. These data well fit with those of Burlina et al., due to the fact that the rate of cesarean section (without labor) was significantly higher in the GDM group than in the control ones, and explains the higher abundance of HSP70 in the placental tissues of the control group.

Liu et al. [24], in a well-conducted, interesting study, investigated the proteome changes of GDM placenta villi to gain evidence for pathway(s) associated with GDM pathogenesis and/or placental remodeling in GDM. For this aim, two-dimensional electrophoresis (2DE) was employed to separate and compare placental protein levels from GDM and normal glucose tolerant pregnant (NGT) groups. To obtain the structural identification of the proteins differentially expressed between the two groups MALDI-TOF/TOF mass spectrometry was employed, so as to gain information on their molecular weight (by MALDI) and amino acid sequence (by TOF/TOF). The data obtained were further confirmed by Western blotting: real time RT-PCR was employed to measure the mRNA levels of the related proteins, while examining the cellular location of the proteins expressed in placenta villi immunohistochemistry (IHC) was performed. These approaches allowed us to identify twenty-one protein spots differentially expressed between GDM and NGT placenta villi in the samples evaluated, and mass spectrometric measurements led to the structural identification of fifteen of them. The molecular functions of these differentially expressed proteins have been discussed by Liu et al. [24]: both mass spectrometric and Western blotting data indicate that the levels of Annexin A2, Annexin A5 and 14-3-3 protein zeta/delta were up-regulated, while the expression of the Ras-related protein Rap1A was down-regulated in the GDM placenta group. These proteins are associated with the development of insulin resistance, transplacental glucose transport, hyperglucose-mediated coagulation and fibrinolysis disorders in the GDM placenta villi.

Sun et al. [25] investigated the differentially expressed proteins from syncytiotrophoblast for severe early-onset preeclampsia in women with GDM. They employed tandem mass tag (TMT) quantitative proteomics, i.e., by using a chemical label that facilitates sample multiplexing in MS-based quantification and identification of biological macromolecules. Previous studies have shown that women with GDM have an increased risk of developing preeclampsia (PE) [26], as the abnormal lipid metabolism caused by GDM leads to the dysfunction of vascular endothelial cells and atherosclerosis, resulting in the onset of PE. However, studies focusing on the pathogenesis of PE in syncytiotrophoblast of GDM patients are lacking. The study of Sun et al. [25] aimed to compare differentially expressed proteins from syncytiotrophoblast between women with GDM and women with GDM that subsequently developed PE. To explore the protein expression changes of syncytiotrophoblast (obtained from pregnant women immediately after delivery) that might explain the pathogenesis of PE in women with GDM, quantitative proteomics was performed using TMT isobaric tags and LC-MS/MS. To investigate the biological processes in which these differentially expressed proteins were involved, bioinformatics analysis was performed. On the basis of the 23 differentially expressed proteins between women with GDM and women with GDM that subsequently developed PE, bioinformatic results showed that the onset of PE in pregnant women with GDM is a multifactorial disorder: the increase of oxidative stress in syncytiotrophoblast, the inadequacy of endometrium infiltration, and the development of angiogenic disorders, could be considered potential mechanisms determining the development of severe early-onset PE in GDM pregnant women. Due to the scarce number of patients evaluated, the results need to be confirmed for a large number of cases.

The aim of the investigation of Szabo et al. [27], based on LC/MS, was to identify PP1, PP8, and PP22 proteins and their placental and trophoblastic expression patterns to evaluate their possible involvement in pregnancy complications. Mass spectrometric data allowed us to identify PP1 as nicotinate-nucleotide pyrophosphorylase (QPRTase), PP8 as ‘serpin B6’ and PP22 as protein disulfide-isomerase. The placenta samples were classified in the following groups: preterm controls, early-onset preeclampsia, early onset preeclampsia with hemolysis, elevated liver enzymes, low platelet count (HELLP) syndrome, term controls, and late-onset preeclampsia. Placentas were used for tissue microarray construction, immunostaining with anti-PP1, anti-PP5, anti-PP8, or anti-PP22 antibodies, and immunoscoring. The PP1 immunoscore was higher in late-onset preeclampsia, while the PP5 immunoscore was higher in early-onset preeclampsia. This behavior reflects the different pathophysiological pathways in these preeclampsia subsets: PP5 dysregulation in early-onset preeclampsia is associated with placental disease, while PP1 dysregulation in late-onset preeclampsia may represent maternal inflammation-inducing placental functional alterations in late-onset preeclampsia.

Assi et al. [28] investigated the placental proteome abnormalities in women with GDM and large-for-gestational-age (LGA) newborns. It is well known that GDM is complicated by short- and long-term newborn complications, among which the occurrence of LGA babies is the more frequent. It is noted that the mechanisms involved in the impaired fetal growth in GDM are not fully understood. Therefore, a proteomic approach was used to analyze placental samples obtained from healthy pregnant women (*n* = 5), patients with GDM (*n* = 12) and with GDM and LGA (*n* = 5). The effects of altered proteins on fetal development were tested in vitro in human embryonic stem cells (hESCs). In pregnant women with GDM and LGA babies, the placental proteome was found to be altered (at least 37 proteins were found differentially expressed as compared with those found in GDM but without LGA babies). In particular, 10 proteins are involved in the regulation of tissue differentiation and/or fetal growth and development. Bone marrow proteoglycan (PRG2) and dipeptidyl peptidase-4 (DPP-4) were highly expressed. Both of these proteins are able to impair the transcriptome profile of stem cells’ differentiation markers when tested in vitro in hESCs, suggesting a potential role in the determination of fetal growth impairment. These interesting results, however, must be confirmed by evaluating a large number of patients with a proper selection of cases and controls. In fact, GDM women were more obese with respect to normal control ones, so it is not clear what the impact is of GDM “per se” and that of obesity.

As a summary of the obtained results, the data obtained by different mass spectrometric approaches, the protein structural identification and their physiopathological roles are reported in Table 1 and Table 2.

The modifications in placenta metabolism and uptake of nutrients can contribute to the long-term complications of neonate of GDM patients [29]. As an example, recent studies have evidenced that some adipokines secreted by adipose tissue (leptin, adiponectin) can impair the uptake of nutrient in the placenta [30]. Recently, attention has been focused on the role of exosomes, nanovesicles that contain proteins lipidis, mRNA, involved in the interorgan communication and whose composition changes in a series of pathological state as hypoxia and obesity [31]. In a study analyzing the exosomes derived by adipose tissue in diabetics with and without obesity, a differential expression of exosomes proteins has been reported [32]. Jayabalan et al. [33] in a very elegant study have analyzed, by mass spectrometry, the exosomes isolated from omental adipose tissue from NGT pregnant women (*n* = 65) and pregnant women with GDM (*n* = 82). The effect of exosomes on human placental cells has been determined using a Human Glucose Metabolism RT2 Profiler PCR array. The result of the study shows that the number of exosomes (vesicles/μg of tissue/24 h) was higher (1.7-fold) in pregnant women affected by GDM with respect to normal pregnant ones. Interestingly, a positive correlation was found between the babies’ birthweight and the number of exosomes. The pathway analysis of the exosomal proteins evidenced differential expression of the proteins targeting the sirtuin signaling pathway, oxidative phosphorylation, and mechanistic target of the rapamycin signaling pathway in GDM compared with NGT. Therefore, these data suggest that in GDM patients, the exosomes of the adipose tissue can modulate the genes involved in the metabolism of glucose of the placenta and the fetal growth.

## 3. Conclusions

The data obtained until now, and described above, briefly described how the newest, different mass spectrometric approaches are a valid tool to investigate the possible changes of placenta physiopathology. The results described above show that quite different results have been obtained in different studies of protein profiles present in placenta samples from healthy and GDM subjects. This can be due, in our opinion, to three different aspects: (i) placental tissue sampling; (ii) instrumental approaches employed for protein profile analysis; (iii) protein identification. In the first case, the presence of large quantities of blood hemoglobin makes it difficult to identify proteins present in low concentrations. In the second case, it must be taken into account that different mass spectrometric approaches reflect on different ionization yields of different proteins, due to the different ionization phenomena present in the ionization methods (e.g., MALDI, ESI) employed in the investigation. Only MALDI was successfully employed to perform ion imaging experiments on both maternal and fetal sides of placenta samples from healthy and GDM subjects. Furthermore, overall, in the case of the direct analysis of the samples obtained from placenta extracts, the complexity of the protein mixture reflects on “matrix effects”, i.e., the privileged ionization of some molecular species with respect to the others. To overcome this behavior, the use of a separative method (either liquid chromatography or 2D gel electrophoresis) prior to the mass spectrometric analysis becomes essential. Finally, for protein identification, the mass spectrometric methodologies as collisional experiments (MS/MS and MS^E^) of the enzymatic digestion products proved to be the most effective methods to be employed for further structural assignments compared with databases. 2D gel electrophoresis and liquid chromatography can lead to effective separation of the different hydrosoluble proteins obtained from the placenta sample, which can be structurally characterized by mass spectrometric analysis of their enzymatic digestion products and their comparison with protein reference databases. It is to be expected that in the future, these approaches will be more specific, with the use of accurate mass measurements obtained by ultrahigh resolution mass spectrometers and by fully automated MS/MS methods. This will give more reliable results to be easily compared among different research groups, so to individuate the metabolic changes in presence of GDM and to implement prevention strategies of maternal and fetal complications and, if possible, to develop suitable pharmacological treatment. However, to reach this aim, studies with a large number of patients and with a correct selection of patients need to be undertaken.

Furthermore, the new single-cell proteomics technology has been applied to the study of placenta, showing promising results in terms of more in-depth knowledge of heterogeneity and molecular interactions of the various placenta cells in the placenta of normal pregnant women [34,35]. In a study evaluating single-cell resolution in placenta from normal pregnant women and GDM, a significant increase of NK and citotoxic T cells and decrease of inflammatory response cells was found in GDM placentas [36]. More studies are necessary to confirm these preliminary data in order to be confident in the clinical applications of this promising technology.

## Figures and Tables

**Figure 1 biomedicines-10-02272-f001:**
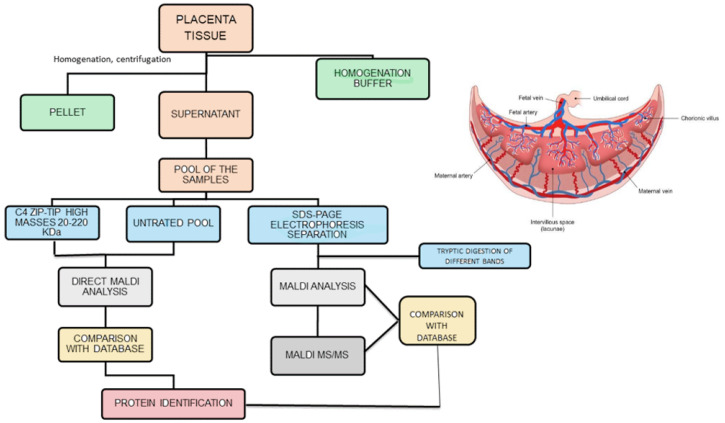
Analytical strategies employed to analyze placenta samples (homogenate was centrifuged twice at 15,000 rpm).

**Table 1 biomedicines-10-02272-t001:** List of proteins upregulated in placenta samples of women affected by GDM and their possible role.

PROTEIN DETECTED	NCBI	PROTEIN ROLE	REF.
Serine/arginine repetitive matrix protein 1	SRRM1	Transcription factor involved in pre mRNA proccessing	[13]
NADH dehydrogenase iron-sulfur protein 3 mitochondrial	NDUFS3	Core subunit of mitochondrial complex 1	[16]
Transcription cofactor vestigial-like protein 4	Vgll4	Nuclear transcription factor	[16]
Voltage dependent anion-selective channel protein 2	VDAC2	Channel through mitochondrial membrane allows diffusion of small molecules	[16]
Galectin-1	Gal-1	Regulator T-cell homeostasis and survival, T-cell immune disorders, inflammation	[19,20,21]
Bcl2-associated transcriptor factor1	BCLAF1	Transcriptional repressor, its over expression induces apoptosis	[13]
Collagen alpha 2 VI chain	COL6A2	Human collagen	[19]
Collagen alpha-1	COLA1	Extracellular matrix collagen	[20,21]
Actin related protein 2 3 complex subunit 2	ARPCS	Protein involved in cell migration and invasion	[19]
Phosphatidylcholine transfer protein	PCTP	Member of the steroidogenic acute regulatory protein-related transfer superfamily.	[19]
Voltage dependent anion selective channel protein 2 2	VDAC22	Transporter of ions and metabolites across the mitochondrial membrane	[19]
Periostin	POSTN	Matricellular protein involved in regulation of cell adhesion, cell differentiation, and organization of extracellular matrix.	[19]
Ig gamma-2 chain C region	IGHG2	Constant region of immunoglobulin heavy chain	[19]
Moesin	MSN	Cytoskeletal adaptor protein that plays an important role in modification of the actin cytoskeleton.	[19]
Triosephosphate isomerase	TPI	Enzyme that catalyzes the reversible interconversion of the triose phosphate isomers dihydroxyacetone phosphate and D-glyceraldehyde 3-phosphate.	[16,19]
Protein disulfide-isomerase	PDI	Enzymes that catalyze exchange reactions between -S-H and S-S- groups, the formation or breaking of sulfur bridges	[19]
Vimentin	VIM	Filament proteinresponsible for maintaining cell shape, integrity of the cytoplasm, and stabilizing cytoskeletal interactions	[19]
14-3-3 protein beta/alpha	GB	Key regulators of processes of mitosis and apoptosis.	[19]
Guanine nucleotide-binding protein G(I)/G(S)/G(T) subunit beta-2	GNB2	Transducer	[24]
26S proteasome non-ATPase regulatory subunit 14	PSDE	Hydrolase Metalloprotease Protease	[24]
Annexin A5	A5	Blood coagulation Signal transduction	[24]
Annexin A4	A4	Anti-apoptosis Signal transduction	[24]
14-3-3 protein zeta/delta	YWHAZ	Anti-apoptosis Signal transduction	[24]
Glycyl-tRNA synthetase	GARS	ATP binding	[24]
Myosin regulatory light polypeptide 9	MYL 9	Calcium ion binding	[24]
Annexin A2	A2	Calcium-dependent phospholipid binding	[24]
Fumarate hydratase mitochondrial	FHM	Fumarate hydratase activity	[24]
Laminin subunit gamma 3	LAMC3	Extracellular marix protein	[19]
L lactate deydrogenase B chain	LDHB	Dehydrogenase protein	[19]
Pyridoxal kinase	PLK	Kinase involved in vitamin B metabolism	[19]
Proteasome subunit beta type 4	PSMB4	Subunit of proteasome involved in cellular proteolisis	[19]
Complement C4A	CG4	Component of human complement	[19]
Trifuncional enzyme subunit alpha mitocondrial	TP	Catalizes the last three of the four reactions of the mitochondrial beta-oxidation pathway	[19]
Calcium binding mitochondrial carrier protein Aralar 2	CMC2	Catalyzes the calcium dependent exchange of cytoplasmic glutamate with mitochondrial aspartate across the mitochondrial membrane	[19]
Csh-like fibrillar surface protein	CSH	Fibrillar surface protein	[19]
Bone marrow proteoglycan	PGR2	Placenta regulation of peptide hormone and growth factor activity	[28]
Amiloride-sensitive amine oxidase	DAO2	Control of histamine levels	[28]
Glia-derived nexin	SERPINE 2	Placental remodelling	[28]
Complement factor H-related protein 1	CFHR1	Complement regulation and lipid metabolism	[28]
Dypeptidyl peptidase 4	DPP49	Regulation of fetal insulin levels and beta-cell development	[28]
Fibronectin	FN1	Placental inflammation	[28]

**Table 2 biomedicines-10-02272-t002:** List of proteins downregulated in placenta samples of women affected by GDM and their possible role.

PROTEIN DETECTED	NCBI	PROTEIN ROLE	REF.
Heath Shock protein	HSP70	Stress-induced protein	[20]
Collagen alpha-1	COLA1	Extracellular matrix collagen	[20,21]
Alpha-1-antitripsin	A1AT	Protease inhibitor	[19]
Alpha-2-antiplasmin	ά2AP	Serine protease inhibitor of plasmine	[19]
ApolipoproteinA-I	APOA-1	Protein component of liporotein	[19]
Clusterin	CLU	Protein associated with the clearance of cellular debris and apoptosis.	[19]
Fibrinogen alpha chain	FGA	Alpha chain of fibrinogen, involved in platelet activation	[19,24]
Fibrinogen beta chain	FGB	Beta chain of fibrinogen, involved in platelet activation	[19,24]
Fibrinogen gamma chain	FGG	Gamma chain of fibrinogen, involved in platelet activation	[19]
Ig gamma-1 chain C region	Ighg1	Constant region of immunoglobulin heavy chains.	[19]
Vitronectin	VN	cell adhesion proteins in plasma.	[19]
Ras-related protein Rap-1A	RAP1A	Signal transduction	[24]
Tryptophanyl-tRNA synthetase citoplasmic	WRS	Triptophan-Trna ligase activity	[24]
NADH-cytocrome b5 reductase 3	CYB5R3	Lipid metabolism	[28]
Corticosteroid 11-beta-dehydrogenase isozyme 2	HSD11B2	Regulation of feto-maternal cortisol levels	[28]
Caspase-1	CASP1	Activation of placental inflammasome	[28]
N(G),N(G)-dimethylarginine dimethylaminohydrolase 1	DDAH1	ADMA_NO pathway homeostasis during pregnancy	[28]

## Data Availability

Not applicable.

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
