# Peer review of "Proteomic Approaches in the Study of Placenta of Pregnancy Complicated by Gestational Diabetes Mellitus"

_biomedicines, 2022, doi:10.3390/biomedicines10092272_

Round 1

Reviewer 1 Report

Title: Proteomic approaches in the study of placenta of diabetic pregnancy.

The manuscript provides an overview of studies that characterized proteomic profiles of placenta from individuals with gestational diabetes mellitus (GDM). The placenta is an important organ that facilitates gaseous, nutrients and waste exchange while serving as a maternal-fetal barrier to block the entrance of substances that could be detrimental to fetal development. The onset of GDM may disrupt the functionality of a placenta, leading to profound implication to the mother and fetus, which highlight the importance to study and review evidence on this topic. However, the way how the information is summarized in the review makes it hard to comprehend. The focus and objective of the review are unclear. Major restructuring is necessary for it to become publishable.

Major comments.

1.    Major restructuring is necessary to improve the readability of the review. The “Placenta protein profile” section covers 6.5 pages. Each cited study occupies about one to three paragraphs, independently. Other than differences in technical approaches, it is unclear to what extent the cited studies agree/disagree with each other. It is recommended to discuss the similarities and differences between different studies in detail. Separating the discussion into smaller sections according to the scope, i.e. proteomic profiles, methodology, biological implications etc. will significantly improve the readability.     

2.    The objective of the review is unclear. The abstract gives the impression that the review will emphasize on different proteomic profiling techniques and highlight their respective strengths and shortcomings. However, the main body is more biologically oriented and highlights different placental proteins enriched in GDM patients across different studies. Yet, in the conclusion, the authors reverted to the technical aspects and highlighted the importance of different proteomic approaches in the field. The switch of scope is quite disruptive to the flow of the review. The lack of objective statement makes it worse for the readers to know which aspects to focus on.

3.    The review could be methodological or biological-oriented, but the implications in both scenarios are not well-discussed. If the review is intended to discuss the development of proteomic approaches to profile placental protein in GDM, emphasis should be given to the superiority of proteomics over other omics approaches, i.e. transcriptomics etc. The advancement of proteomic technology to mitigate the gaps in protein identification and specificity and sensitivity should also be elaborated. On the other hand, if the review is biologically inclined, it should discuss the significance of GDM placental proteomes in diagnosis, pathophysiology and prognosis of long-term health impacts to both the mother and fetus. Without the meaningful implication in their respective context, it is hard for the readers to appreciate the importance of this manuscript.

4.    The authors should also comment on the gaps in existing research paradigms. This again, depends on the focus of the review. If it is methodologically oriented, what improvement can be done to improve the proteomic strategies for placental protein profiling? What could the new single-cell proteomics technology bring to the table? If it is biologically oriented, what are the exploitable target pathways? Are there any additional value that the current research fails to identify? The authors also concluded on the possibility to develop suitable pharmacological treatments with the data. What would be the disease to treat and how? Please elaborate. It is also good to pinpoint the limitation of existing studies.   

5.    Grammar and language are an issue. Spelling mistakes are common throughout the entire manuscript. Getting professional editing service is suggested.

Reviewer 2 Report

With great pleasure, I have read the article entitled: Proteomic approaches in the study of placenta of diabetic 2 pregnancy. The authors have taken into consideration the important problem: gestational diabetes mellitus. From a future generation point of view, it is an important problem, due to the fact that diabetes, especially type II, has been noted as a civilization disease. However, I have one critical remark which forces the lack of publication recommendation i.e: the lack of medical justification of the discussed problem which is the clue of the article. Therefore the medical, and physiological consequences of GDM should be strongly pointed out in the text.

Reviewer 3 Report

The review of Lapolla and Traldi evaluates the potential that proteomics can offer through the analysis of the placenta of women with gestational diabetes.

This innovative approach, based on the qualitative and quantitative evaluation of proteins, could provide new information on patients with gestational diabetes. 

Title. the term “diabetic pregnancy” also could include pregnancies of women with pre-gestational diabetes. Could it be appropriate to specify it also in the title?

Figure 1. Is it possible to insert some images / colors in figure 1? E.g.image of the placenta. Moreover, it might be useful to also enter some information such as homogenate was centrifuged twice 69 at 15,000 rpm for 10 min

Table 1 reports the proteins detected or present in abundance in the placenta of women with gdm.

Is it possible to split table 1 into two tables? in one you could insert the proteins present in abundance and in the other those only detected.

furthermore, it could be useful to indicate the differences with the placenta of women without gdm

Round 2

Reviewer 1 Report

This is a revised manuscript summarising the proteomic profiles of placenta of individals with gestational diabetes mellitus (GDM).

1.    The objective, which is to discuss the results obtained by different mass spectrometric approaches in the study of placental protein profile, is now clearly specified in the abstract. Of course, the review has provided a comprehensive overview of the existing proteomic approaches used to study placental proteins (the appropriateness of presentation format is arguable; see Comment No.2). What is truly lacking the discussion and comparison between different mass spectrometric approaches. After reading the review, I failed to capture the advantages and disadvantages of different approaches, i.e. MALDI, 2DE LC-MS/MS, label-free LC-MS or MALDI-TOF/TOF (or they are equally good?). Each of them seems to give different profiles (based on Table 1 and 2), but why? It was also mentioned in the review that different placental sides resulted in different profiles, but what is the underlying reasons? Which placental side should be used in future experiments? These are just some aspects that can be discussed in further details. Essentially, the review provides a good compilation of evidence that the mass spectrometry technology can be used to profile placental proteins (which is already well-established considering that the technology but fails to highlight the gaps and progressive advancements of the research topic. More critical inspection and discussion of the existing evidence is needed.

2.    The “placenta protein profile” section has been revised and condensed significantly. While the readability has been improved, this section still reads very much like a compilation of summaries from different studies, instead of a critical appraisal of the research topic. This is largely attributable to the strategy taken, i.e. samples analyzed – instrumental approaches employed – obtained results – physiopathological meaning of the results. Considering the repetitiveness of the content which the authors agree, these points could be better presented in a Table. By doing so, the main text can be used for in-depth discussion of methodological and/or biological content.

Author Response

First of all we wish to thank the editor and the reviewer for their carefula evaluation of the manuscript. We have taken into account their comments and incorporated them in the manuscript. Then it must be stressed that the manuscript follows the rules indicated by the editor pertaining the lenght and the number of tables and figures.

The advantages and disadvantages of the different approaches have been specified in the text (in yellow), (see lines 19-20, 78-81,102-104,120-122, 140-141, 231, 275-276, 307-311, 355-375,384-386).

The reasons determining different results analyzing different placenta sides and the specification on the preferential placenta side to be utlized has been added (see lines 151-153, 159-164).

Reviewer 2 Report

After correction, I can recommend the sent article for publication.

Author Response

the medical and physiological consequences of GDM have been deelpy desribed in the introduction sectio (page 1 )